# Graph Spectral Regularization for Neural Network Interpretability

## Abstract

Deep neural networks can learn meaningful representations of data. However, these representations are hard to interpret. For example, visualizing a latent layer is generally only possible for at most three dimensions. Neural networks are able to learn and benefit from much higher dimensional representations but these are not visually interpretable because nodes have arbitrary ordering within a layer. Here, we utilize the ability of the human observer to identify patterns in structured representations to visualize higher dimensions. To do so, we propose a class of regularizations we call *Graph Spectral Regularizations* that impose graph-structure on latent layers. This is achieved by treating activations as signals on a predefined graph and constraining those activations using graph filters, such as low pass and wavelet-like filters. This framework allows for any kind of graph as well as filter to achieve a wide range of structured regularizations depending on the inference needs of the data. First, we show a synthetic example that the graph-structured layer can reveal topological features of the data. Next, we show that a smoothing regularization can impose semantically consistent ordering of nodes when applied to capsule nets. Further, we show that the graph-structured layer, using wavelet-like spatially localized filters, can form localized receptive fields for improved image and biomedical data interpretation. In other words, the mapping between latent layer, neurons and the output space becomes clear due to the localization of the activations. Finally, we show that when structured as a grid, the representations create coherent images that allow for image-processing techniques such as convolutions.

## 1 Introduction

Neural networks have revolutionized many areas of machine learning including image and natural language processing. However, one of the major challenges for neural networks is that they are still black boxes to the user. It is not quite clear how network internals map from inputs to outputs or how to interpret the features learned by the nodes. This is mainly because the features are not constrained to have specific structure or characteristics. Existing regularizations constrain the learned code to have certain properties. However, they are not designed to specifically aid in interpretation of the latent encoding. For example, $L_1$ regularization induces sparsity in the activations but does not impose specific structure between dimensions.

Here, we introduce a new class of regularizations called *Graph Spectral Regularizations* that result in activations that are filtered on a predefined graph. We define specific members of this class for applications. First, we introduce a (graph) *Laplacian smoothing regularization* which enforces smoothly varying activations while at the same time reconstructing the data. This regularization is useful for learning features with specific topologies. For instance, we show that on a cluster-structured topology where features correspond to hierarchical cluster structure in the data it reflects the abstract grouping of features. We also show it is useful for inducing feature consistency between nodes of capsule networks (Sabour et al., 2017). The graph regularization semantically aligns the features such that they appear in the same order in each capsule. When trained on MNIST digits, we find that each of our 10 capsules consisting of 16 nodes encodes the same transformation (rotation, scale, skew, etc) of a particular digit in the same node.

While the Laplacian smoothing regularizations is useful in the context where the features of the data have a recognizable topology, often we don't know the explicit structure of the data. Instead, we would like to extract the topology of the data itself. Thus, we design a filter that encourages the graph structure layer to learn data-shape features. We achieve this by using a spatially localized, Gaussian filter to localize the activations for any particular data point. We ensure that only one of a dictionary of localized filters is chosen as the activation via a *spectral bottleneck* layer preceding the graph-structured layer. We show that spatially-localized filter regularizations are useful for detecting circular and linear topologies of data that are not immediately reflected by the observed features. We also explore a biological system – a single-cell protein expression dataset depicting T cell development in the thymus – that has continuous progression structure. The graph structured layer (with a ring graph) reveals the data to have a Y-shaped topology reflecting the bifurcation into CD4 (regulatory) and CD8 (cytotoxic) T cells, confirming known T cell biology.

Finally, we show that the graph-structured layer, when imposing a 2D grid, creates a "pseudo" image that can be analyzed by convolution layers. We show that such re-encoded images of MNIST digits have localized receptive fields that can be used for classification and visual interpretability. Interestingly, we find that the convolution obviates the need for a spectral bottleneck as the convolution and max pooling themselves may provide that function.

Our contributions are as follows:

- A framework for imposing graph structure on latent layers using graph spectral regularizations.
- A Laplacian graph smoothing regularization and its applications in learning feature smoothness and consistency.
- Spatially localized graph regularizations using a spectral bottleneck based on a dictionary of Gaussian Kernels and its application in recognizing data topology.
- Applications of graph spectral regularizations, natural and biological datasets to demonstrate feature interpretability and data topology.

The rest of this paper is organized as follows. We first define graph structured layers and two techniques utilizing this layer in Section 2. Then we present experiments demonstrating improved interpretability in Section 3. Finally, we wrap up with conclusions in Section 4.

## 2 GRAPH-STRUCTURED LAYER

Consider a given layer in an artificial neural network, and let $\mathcal{V} = \{\nu_1, \ldots, \nu_n\}$ be the neurons in this layer. These neurons can essentially be regarded as functions of inputs to the neural network, and map each input to an $n$ dimensional vector $\boldsymbol{\nu}(x) = [\nu_1(x), \ldots, \nu_n(x)]^T \in \mathbb{R}^n$. Typically, no particular structure is being directly imposed on the range of $\boldsymbol{\nu}$ beyond general notions of bounded $L_p$ norm (e.g., Euclidean $L_2$ norm or $L_1$ norm as a proxy for sparsity). This is clear with fully connected layers, but even with convolutional ones that introduce some relation between neurons within each layer, there is still no clear topological structure for the representations obtained by the entire layer as a whole. Indeed, convolutional layers typically learn multiple channels (or filters) from input signals, and while neurons within each channel can be organized in spatiotemporal coordinates, there are no imposed relations between the different channels.

Since many applications of neural networks essentially focus on supervised learning tasks (whether predictive or generative), where hidden layer neurons are only used as intermediate computational units, their unstructured nature is not considered an important issue. However, when using neural networks for unsupervised and exploratory tasks, in which the hidden layers are treated as latent data representations, their lack of structure makes the interpretation of the resulting representations challenging if not impossible. To address this challenge, autoencoders and similar unsupervised deep models typically restrict their bottleneck layer to have two or three neurons, thus mapping data points to $\mathbb{R}^2$ or $\mathbb{R}^3$, where the entire data can be visualized as a 2D or 3D cloud of points. While this approach is useful for getting a general understanding of structures (e.g., clustering or trends) in the data, it significantly limits the amount of information that can be captured by such latent representations. Furthermore, it underutilizes the ability of a human observer to identify rich patterns in 2D displays. Indeed, human perception in natural settings is not tuned to observe particle clouds,

but rather to recognize shapes, identify textures, and assess relative sizes of objects. Such elements are often used by visualization techniques in various fields, such as TreeMap (Shneiderman, 1992), audio spectrograms (Flanagan, 1972), and even classic box plots and histograms (Tukey, 1977).

Here, we propose a new approach for producing human interpretable patterns in the latent representation obtained by hidden layers of neural networks. To this end, we impose a graph topology on the neurons in $\mathcal{V}$, which enables us to control spectral properties (e.g., regularity and locality) of latent representations obtained by them to produce recognizable patterns. Formally, we consider the neurons $\mathcal{V}$ as vertices of a graph with weighted edges between them defined by an $n \times n$ adjacency matrix $W$. Then, for any given input $x$ to the network, we consider the activations $[\nu_1(x), \ldots, \nu_n(x)]$ as a signal $\boldsymbol{\nu}$ over this neuron graph. We propose here three ways to utilize this graph signal structure of a neural network layer. First, by considering graph signal processing notions, we can define spectral regularizations that control the regularity or smoothness of learned signals, as explained in Section 2.1. Second, by utilizing sparse graph wavelet dictionaries, we can define a new type of bottleneck that forces autoencoder middle layers to encode inputs by sparse set of localized dictionary atoms (e.g., as done in compressive sensing and dictionary learning), as explained in Section 2.2. Finally, by utilizing simple graph structures, such as a ring graph or a 2D lattice we can force downstream layers to be convolutional layers, regardless of whether the input of the network was structured or not. Section 3 demonstrates the utility of each of these design choices for several data exploration applications, in both supervised and unsupervised settings.

## 2.1 SPECTRAL GRAPH REGULARIZATION

The graph structure defined by the weighted adjacency matrix $W$ naturally provides a notion of locality, based on local graph neighborhoods, or geosesic distances on the graph. However, recent works on graph signal processing (e.g., Shuman et al., 2013, and references therein) also explored spectral notions provided by graph structures, which extend and generalize traditional signal processing and harmonic analysis notions. These notions are based on the definition of a graph Laplacian as

$$\mathbf{L} = D - W, \quad D = \mathrm{diag}(\deg(\nu_1), \ldots, \deg(\nu_n)), \quad \deg(\nu_j) = \|W_{(j,\cdot)}\|_1, j = 1, \ldots, n,$$

which provides (either directly or via proper normalization) a discrete version of well-studied manifold Laplace operators from geometric harmonic analysis (e.g., Belkin & Niyogi, 2002; Coifman & Lafon, 2006). Then, the eigenvectors $\psi_0, \ldots, \psi_{n-1}$ of $\mathbf{L}$ (indexed by convention in ascending order of the corresponding eigenvalues) are considered graph Fourier harmonics. These can be shown to converge to discrete Fourier harmonics when considering a ring graph, in which case their associated eigenvalues, $0 = \lambda_0 \leq \cdots \leq \lambda_{n-1}$, correspond to squared frequencies. This understanding gives rise to the graph Fourier transform defined as a linear transformation characterized by the matrix $\Psi$ whose columns are the graph harmonics $\{\psi_j\}_{j=0}^{n-1}$. When applied to a neuron-activation signal $\boldsymbol{\nu}$ in our setting we get its graph Fourier coefficients as $\widehat{\boldsymbol{\nu}} = \Psi^T \boldsymbol{\nu}$, where $\widehat{\boldsymbol{\nu}}$ is associated with the graph (squared) frequency $\lambda_j$. Similarly, the inverse Fourier transform is also defined via $\Psi$, as it can be verified that $\Psi\widehat{\boldsymbol{\nu}} = \Psi\Psi^T\boldsymbol{\nu} = \boldsymbol{\nu}$, since the graph Laplacian (for all graphs considered in this work) yields a full orthonormal set of eigenvectors, and thus $\Psi$ is an orthogonal matrix.

The graph Fourier transform allows us to consider the activations $\boldsymbol{\nu}(x)$ for a given input $x$ of the network in two domains. First, in their original form, these activations form a signal over the neuron graph, which can essentially be considered as a function of individual neurons, with $\boldsymbol{\nu}_{(x)}[i] = \nu_i(x)$, $i = 1, \ldots, n$, which we refer to as the neuron-domain representation. Alternatively, we can consider this signal in the spectral domain, via its Fourier coefficients, as a function $\widehat{\boldsymbol{\nu}}_{(x)}[j], j = 0, \ldots, n-1$, of graph-harmonic indices. This allows us to pose a new set of regularizations that are defined in the spectral domain, rather than in the neuron domain, in order to directly enforce spectral properties of the neuron activation signal.

We note that one of the most popular regularization traditionally used in deep learning is the $L_2$ regularization, which essentially adds the squared Euclidean norm of the activations in the neuron domain (i.e., $\|\boldsymbol{\nu}(x)\|_2^2$) as another term in the main optimization target for gradient descent. Such regularization encourages the activations to all have equivalently small values, essentially providing a global notion of smoothness that is then balanced with other loss terms but typically provides stability to the optimization process (Goodfellow et al., 2016). Since it can be verified that the graph Fourier transform is energy preserving (i.e., due to the orthonormality of graph harmonics), this

regularization can equivalently be considered in the spectral domain by using $\|\widehat{\boldsymbol{\nu}}(x)\|_2^2$ instead of $\|\boldsymbol{\nu}(x)\|_2^2$. However, unlike the neuron domain, in the spectral domain we can associate with each element in $\widehat{\boldsymbol{\nu}}(x)$ a frequencial interpretation given by $\{\lambda_j\}_{j=0}^n$, and therefore we can also generalize the spectral $L_2$ regularization to be weighted by functions of these squared-frequencies. Namely, given weights $\mu_j = \boldsymbol{\mu}(\lambda_j)$, $j = 0, \ldots, n-1$, for some function $\boldsymbol{\mu}$, we define the spectral $L_2(\mu)$ regularization as adding the term

$$\|\widehat{\boldsymbol{\nu}}(x)\|_{L_2(\boldsymbol{\mu})}^2 = \sum_{j=0}^{n-1} \mu_j \left(\widehat{\boldsymbol{\nu}}_{(x)}[j]\right)^2 = [\widehat{\boldsymbol{\nu}}(x)]^T \boldsymbol{\mu}\, \widehat{\boldsymbol{\nu}}(x),$$

where $\boldsymbol{\mu} = \mathrm{diag}(\mu_0, \ldots, \mu_{n-1})$, to the optimization loss of the neural network. Notice that such weighting cannot be directly defined in the neuron domain, as individual neurons are not associated with any a priori interpretation. Instead, to apply the spectral regularization in the neuron domain, we consider the matrix form on the RHS, and by combining it together with the definition of the graph Fourier transform we can write the $L_2(\mu)$ spectral regularization term in the neuron domain as the quadratic form $[\boldsymbol{\nu}(x)]^T \mathcal{F}(\boldsymbol{\mu})\boldsymbol{\nu}(x)$, where $\mathcal{F}(\boldsymbol{\mu}) = \Psi \boldsymbol{\mu} \Psi^T$ is a matrix that is independent of the network input $x$, and can thus be directly computed in advance from the neuron graph structure, and the predetermined spectral weights in $\boldsymbol{\mu}$. Finally, while in this work we focus on the $L_2(\mu)$ form of spectral regularizations, in general other weighted norms (e.g., $L_1(\mu)$ or more generally $L_p(\mu)$, $p \geq 1$) can also be used to define spectral regularizaions based on based on the harmonic structure induced by the neuron graph.

**Laplacian smoothing regularization:**  we now focus on given class of $L_2(\mu)$ spectral regularizations with nonnegative weights, i.e., $\boldsymbol{\mu} \geq 0$. In such cases, the chosen weights determine which harmonic bands to penalize, and by how much, in the resulting regularization. Therefore, these spectral regularization enable to guide the latent representation provided by the graph-structured layer towards, or away from, certain harmonic patterns. In particular, this enables us to encourage smooth latent representations (over the neuron graph sense) by using weights that penalize high frequency. A natural choice for such weights is to simply set them by the identity $\boldsymbol{\mu}(\lambda_j) = \lambda_j$, which results in $\mathcal{F}(\boldsymbol{\mu}) = \mathbf{L}$. We refer to this regularization as Laplacian smoothing, as the resulting quadratic loss term for it is $[\boldsymbol{\nu}(x)]^T \mathbf{L}\boldsymbol{\nu}(x)$, and demonstrate its utility for producing interpretable latent representations in hidden layers of neural networks in Section 3.

## 2.2  Spectral bottleneck

The spectral regularization defined in Section 2.1 is based on representing and activation signal as a linear combination of graph harmonics, and then formulating a regularization over the corresponding coefficients for the combination, which are given by the graph Fourier transform. This principle can be extended by considering more general notions of dictionary learning (Olshausen & Field, 1997). In general, such methods seek to define a dictionary of representative atoms that can be used to effectively represent signals, while capturing (in each atom) certain patterns, such as spatial or spectral locality. Under this terminology, the graph Fourier transform is based on a dictionary $\{\psi_j\}j = 0^n$ consisting of the graph harmonics as atoms that are extremely local in frequency. However, it can be shown that these atoms are often not spatially local over the neuron graph. To extend our approach to also include notions of spatial locality, we propose to also consider dictionaries that are based on bandlimited filters, such as graph wavelets or translated Gaussians (Hammond et al., 2011). Let $\{\phi_\xi\}_{\xi=1}^\ell \subset \mathbb{R}^n$ be such a dictionary, and let $\Phi$ be a $n \times \ell$ matrix whose columns are the $\ell$ atoms in the dictionary. We note that while in general, the atoms in the dictionary need not be orthonormal, we assume they are chosen such that the matrix $\Phi$ has a suitable pseudoinverse $\Phi^\dagger$ such that $\Phi^\dagger \Phi = Id_{\ell \times \ell}$. Then, the best approximation of a neuron signal $\boldsymbol{\nu}(x)$, for network input $x$, by a linear combination of dictionary atoms is given by $\Phi\Phi^\dagger \boldsymbol{\nu}(x)$, which can be written directly as a linear combination $\sum_{\xi=1}^\ell \varphi_\xi^{\boldsymbol{\nu}(x)} \phi_\xi$ with the dictionary coefficients given by $(\varphi_1^{\boldsymbol{\nu}(x)}, \ldots, \varphi_\ell^{\boldsymbol{\nu}(x)})^T = \boldsymbol{\varphi}(\boldsymbol{\nu}(x)) = \Phi^\dagger \boldsymbol{\nu}(x)$.

The dictionary coefficients computed by $\boldsymbol{\varphi}$ provide a dictionary-based extension of the Fourier coefficients computed by the graph Fourier transform. Therefore, the same spectral regularization discussed in the previous section can be be directly generalized to dictionary-based regularization. However, the dictionary coefficients also provide an alternative utilization as a new type of infor-

mation bottleneck in the neural network, inspired by sparse representations commonly used in compressive sensing (Qaisar et al., 2013). In particular, here we consider a bottleneck that forces the network to project its neuron activation signal $\nu(x)$ on a single dictionary atom, which would be chosen adaptively depending on the input $x$, and then only pass this atom to subsequent layer. To achieve this bottleneck, we split the graph-structured layer into two parts $\nu$ and $\nu'$, before and after the bottleneck (correspondingly), such that

$$\nu'(x) = \Phi\, \boldsymbol{\varphi}_{max}(x), \quad \boldsymbol{\varphi}_{max}(x) = \text{softmax}[\boldsymbol{\varphi}(\nu(x))],$$

where softmax is defined as a function that maps vectors in $\mathbb{R}^n$ to unit $L_1$ norm approximately one-hot vectors (Bishop, 2006). Therefore, previous layers in the network feed into the neuron graph signal $\nu$, which is then passed through the bottleneck to produce a filtered signal $\nu'$ based on approximately one atom out of the $\ell$ provided ones in the dictionary, and then the activations form this new signal are passed on to subsequent layers. In Section 3, we show by using a spectral bottleneck that is based on graph-translated Gaussians as dictionary atoms, we can force individual inputs processed by the network to be projected on local regions of the neuron graph. This, in turn, allows us to organize the latent representation in the graph-structured layer into "receptive fields" over the neuron graph, which capture local regions in the input data and uncover trends in it based on the imposed graph structure.

## 3 EMPIRICAL RESULTS

**Topological Inference Using Laplacian Smoothing Regularization**    First, as a sanity check, we demonstrate graph spectral regularization on data that is generated with a specific topology. Our data has a hierarchical cluster structure, where there are 3 large-scale structures, each comprising two Gaussian subclusters generated in 15 dimensions (See Figure 1). We use a graph-structure layer with 6 nodes with 3 connected node pairs and employ the Laplacian smoothing regularization. After training, we find that each node pair acts as a "supernode" that detects each large scale cluster. Within each supernode, each of the two nodes encodes one of each of the two Gaussian substructures. Thus, this specific graph topology is able to extract the hierarchical topology of the data.

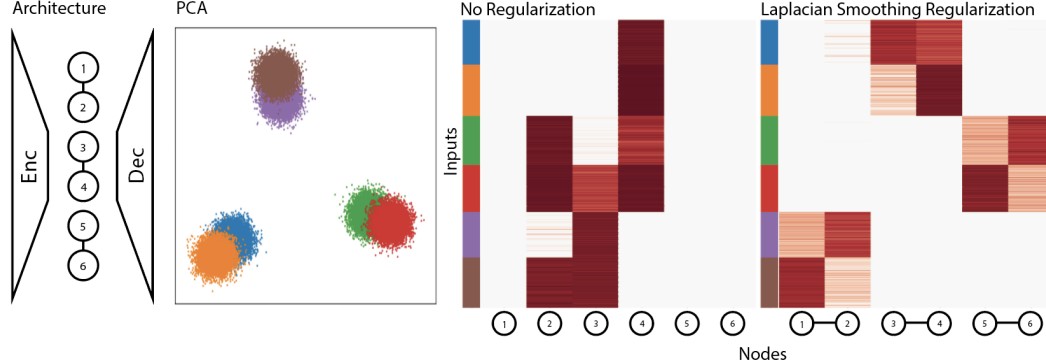

Figure 1: From left to right, network architecture, 2D PCA of artificial data, activation heatmap of a standard autoencoder without regularization, and an activation heatmap of an autoencoder with Laplacian smoothing regularization. We use a graph of six nodes with three pairs of connected nodes. The activation of each pair of nodes corresponds to one of the three clusters, where the differences in activation between the two nodes corresponds to which of the two subclusters the point belongs to. In the model with Laplacian smoothing we are able to clearly decipher the structure of the data. Whereas with the standard autoencoder the overall structure of the data is not clear, for example, the fact that the the blue and orange points belong different subclusters.

**Semantic Feature Organization in Capsule Networks**    Next, we demonstrate Laplacian smoothing regularization on a natural dataset. Here, instead of using an autoencoder framework, we use a capsule network consisting of 10 capsules of 16 nodes. In the original capsule network paper,

Sabour et al. (2017) construct an architecture that is able to represent each digit in a 16 dimensional vector using a reconstruction penalty with a decoder. They notice that some of these 16 dimensions turn out to represent semantically meaningful differences in digit representation such as digit scale, skew, and width. We train the capsule net on the MNIST handwritten digit dataset with the Laplacian smoothing regularization applied between the matching ordinal nodes of each capsule using fully connected graphs. We show in Figure 2 that without the regularization each individual capsule in the network derives its own ordering of features that it learns. However, with the graph regularization we obtain a consistent feature ordering, e.g. node 5 corresponds to line thickness across all digits. Thus, the Laplacian smoothing regularization enforces a more interpretable encoding with "tunable" knobs that can be used to generate data with specific properties, as shown in Figure 2.

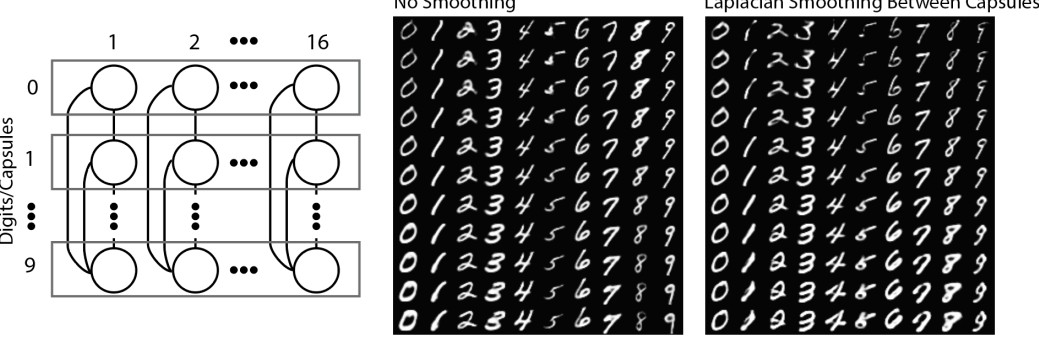

Figure 2: The prediction layer of a capsule network on MNIST uses 10 digit capsules each with 16 dimensions. To perform Laplacian smoothing between capsules we create 16 fully connected graphs of size 10. Next, we show dimension perturbations for each digit. Each columns shows the reconstruction when one of the 16 dimensions in the DigitCaps representation is tweaked by 0.05 in the interval [-0.25, 0.25]. On the left we see a single dimension in a standard capsule net across all digits, and on the right we see a dimension on a capsule net trained with Laplace smoothing regularization. With Laplacian smoothing regularization between capsules (right) a single dimension represents line thickness for all digits. Without this regularization each digit responds differently to perturbation of the same dimension.

**Spectral Bottleneck Regularization**   Next, we impose a linear graph-structure on a dataset generated to have a one-dimensional progression in 20 ambient dimensions (see Figure 3). Here we see that without any graph structure regularization the encoding by the features is arbitrary. Once the Laplacian smoothing regularization is enforced, subsequent points in the progression have smoothly varying changes. Next, in order to make features correspond to data topology, we introduce a spectral bottleneck using an additional layer preceding the spectral regularization layer. This bottleneck layer, using a softmax, effectively chooses one atom from a dictionary of Gaussian kernel-shaped filters for the activations. We see that with the addition of this regularization, we have features of the layer encoding (and activating for) different parts of the graph.

**Topological Analysis of T cell Development**   Next, we show that spatially-localized filter regularizations are useful for learning characteristic features of different parts of the data topology. We test this ability on biological mass cytometry data, which is high dimensional, single-cell protein data, measured on differentiating T cells from the Thymus (Setty et al., 2016). The T cells lie along a bifurcating progression where the cells eventually diverge into two lineages (CD4+ and CD8+). Here, we see that the spectral bottleneck compactly encode the branches in specific nodes and thereby create a receptive field for the data topology. Examination of these nodes reveals the input protein features that characterize the different parts of the progression. From the activation heatmap, we see that one major differences between the blue and green branches is the activation of node 18. We see from the heatmap that correlates nodes to gene activations that cluster 18 is positively correlated CD8 and negatively with CD4, and thus this node is the *switch* between the two lineages. For the cluster of nodes from 6-12, these nodes are low in the red branch and high in the other two branches. Since these nodes are positively correlated with CD3, TCRb and CD127, this indicates

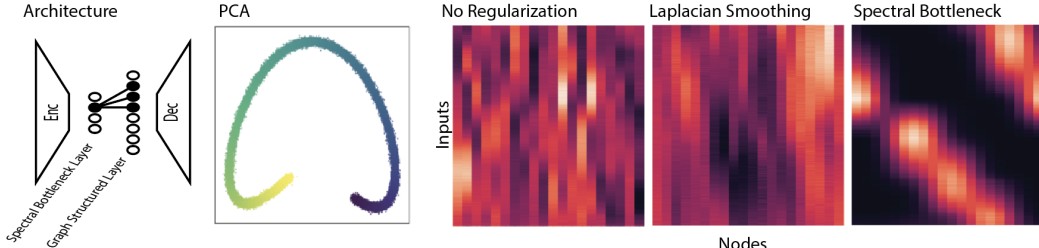

Figure 3: From left to right, Architecture diagram of the spectral bottleneck layer, PCA plot of the data, embedding layer activation heatmaps ordered by location on the generated line with no smoothing, graph smoothing, and spectral bottleneck modifications. It is difficult to identify the linear structure of the data in a standard autoencoder. With Laplacian smoothing in a line graph we start to see a diagonal structure emerge. Finally, adding the spectral bottleneck layer we are able to clearly see the one-dimensional structure of the data even in an embedding space with much higher dimensionality than usually possible.

that nodes further along in differentiation (blue and green branches) indicating that the cells have acquired higher levels of canonically mature T cell markers (CD3 and TCRb) as well as the naive T cell marker CD127. Although this analysis was done on a relatively low-dimensional dataset which could be analyzed using other methods, it corroborates that the receptive fields produced by the spectral bottleneck offer meaningful features that can be examined to characterize parts of the data topology and can be applied to more complex, higher-dimensional datasets.

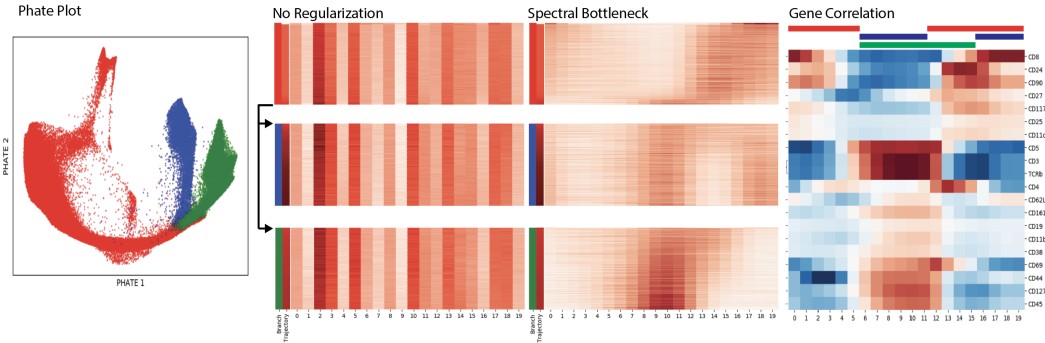

Figure 4: From left to right, shows a 2D embedding (PHATE (Moon et al., 2017)) of the T cell dataset, heatmaps of embedding layer activations sorted by branch and trajectory labels with and without graph spectral regularization, and finally a correlation matrix between the 20 genes and the 20 nodes of the embedding. The data consists of two T Cell developmental trajectories, from the red branch to the blue branch and from the red branch to the green branch. The gene correlation plot depicts blue as negative correlation, white as no correlation, and red as positive correlation. As we can see graph spectral regularization creates an interpretable and biologically relevant embedding, splitting into CD4+ (green) and CD8+ branches (blue).

**Pseudo-Images and Convolutions for Human Interpretable Encodings**    Finally, we show the capability of graph-structured regularizations to create pseudo-images from data. Without graph-structured regularization, activations appear unstructured to the human eye and as a result are hard to interpret (See Figure 5). However, using Laplacian smoothing over a 2D lattice graph we can make this representation more visually distinguishable. Since we now take this embedding as an image, it is possible to use a standard convolutional architecture in subsequent layers in order to further filter the encodings. When we add 3 layers of 3x3 2D convolutions with 2x2 max pooling we see that representations for each digit are compressed into specific areas of the image (Figure 5). Now, by visual inspection of this high dimensional embedding layer, we are able to quickly visually categorize inputs. We show that the layer is segmentable, with receptive fields for each digit, thus

making the layer amenable for classification. Further, we note that the classification penalty along with the graph-structured layer by itself induces spatial localization, without a spectral bottleneck layer or localized filter regularization. We speculate that this is a result of the convolutions combined with max pooling inducing spatially localized features.

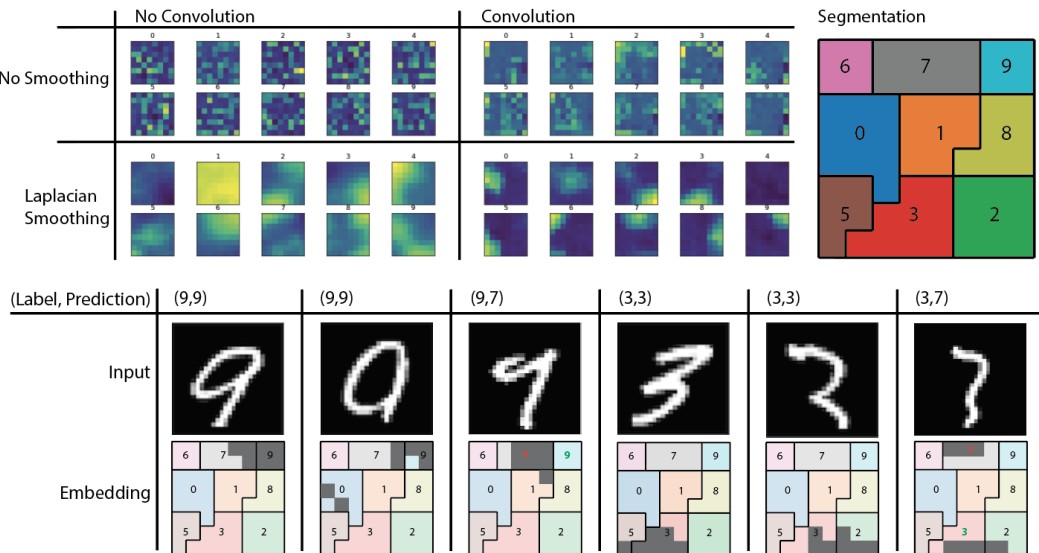

Figure 5: Shows average activation by digit over a 64 (8x8) 2D grid using Laplacian smoothing and convolutions following the regularization layer. Next, we segment the embedding space by class to localize portions of the embedding associated with each class. Notice that the digit 4 here serves as the null case and does not show up in the segmentation. Finally, we show the top 10% activation on the embedding of some sample images. For two digits (9 and 3) we show a normal input, a correctly classified but transitional input, and a misclassified input. By inspection of the embedding space we can see the highlighted regions of the embedding space correlate with the semantic description of the digit type.

## 4    CONCLUSION

Here, we have introduced a class of graph spectral regularizations that impose graph structure on the activations of hidden layers and show they allow for more interpretable encodings. These include a Laplacian smoothing regularization that creates locally smooth activation patterns which can reflect structure and progression in the associated data as well as consistency of features as demonstrated on capsule nets. Next, we show that if we constrain the node activations to be more spatially localized on the imposed graph structure, using wavelet-like filters, we enable the hidden layers to learn features associated with different parts of the data topology. For example, we can extract biologically meaningful characterizations of a bifurcating differentiation structure in mass cytometry data measuring T cell differentiation. Finally, we show that graph structured regularizations can create pseudo-images when the underlying graph is a grid, making the data amenable to convolutions and other image-processing techniques such as segmenting. We show that such segmentation gives receptive fields that allow for human interpretable activations of high dimensional hidden layers. Normally, visualization comes at the cost of dimensionality as only layers containing two or three dimensions can be visualized. Finally, we note that graph structured regularizations encode datapoints as signals on the graph and thus graph signal processing may be used in future work to analyze such data.

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

## APPENDIX A    ARTIFICIAL DATA GENERATION

**Hierarchical Cluster Dataset**    We simulate three gene modules each with two sub modules for a total of six total clusters. There are 15 genes where five are associated with each larger module. For every datapoint exactly one of the three gene modules is "active". This is represented by a mean shift of 10 in all genes associated with the module. To distinguish the sub-modules within larger modules, one of the submodules has an additional meanshift of 10 in two of the five genes when active. We then add gaussian noise independently to each feature with mean zero and standard deviation one.

Figure 6: From left to right, 2D Multidimensional scaling (MDS) plot on the dataset, graph smoothing regularization structure, and heatmaps showing activation strength of datapoints on the y-axis ordered by rotation and nodes on the x-axis. We can see with too much smoothing ($\alpha = 0.1$) we get a less meaningful heatmap. But more reasonable values of $\alpha$ result in activations of lower period both vertically and horizontally.

**Linear Dataset**  We simulate a linear dataset by sequential feature activation. We generate labels $y$ by uniformly sampling numbers between zero and ten, $y \sim \mathcal{U}(0, 10)$. Looking feature by feature, the first feature has values approximately equal to the probability distribution function (pdf) of the normal distribution with mean one and standard deviation one. The second feature has values approximating the pdf of $\mathcal{N}(2, 1)$, and the third feature $\mathcal{N}(3, 1)$. We generate 60,000 data points in this way and add independent gaussian noise with mean zero and standard deviation of 0.001.

**Effect of increasing Laplacian smoothing regularization**  We analyze both Laplacian smoothing and spatially-localized representations on images of a rotating teapot (See Figure 6) (Weinberger et al., 2004). We include a graph spectral layer with a ring topology of 20 nodes. We gradually increase the smoothness coefficient (from 0 to 0.1) and show that, at intermediate levels of the coefficient (between 0.0001 and 0.001), we obtain an activation pattern that is smooth on the imposed ring graph. At 0 regularization there is no smoothness on the graph topology (horizontal smoothness in the activation heatmap) and the activations appear to be randomly ordered. At intermediate smoothing we can observe smooth structures with seemingly ordered activations. At high smoothing the activations all become the same effectively creating one node/dimension in the activations (at 0.01 and higher).

## APPENDIX B  EXPERIMENT SPECIFICS

We use Leaky relus with a coefficient of 0.2 (Maas et al.)  for all layers except for the embedding and output layers unless otherwise specified. We use the ADAM optimizer with default parameters (Kingma & Ba, 2014).

**Laplacian Smoothing on an Autoencoder**  We use an autoencoder with five fully connected layers. The layers have widths [50,50,20,50,50]. To perform Laplacian smoothing on this autoencoder we add a term to the loss function. Let $\nu$ be the activation vector on the embedding layer, then we add a penalty term $\alpha \nu^T \boldsymbol{L} \nu$ where $\alpha$ is a weighting hyperparameter to the standard mean squared error loss.

**Spectral Bottlenecking in an Autoencoder**  To build a spectral bottleneck layer let $\Phi$ be an $n \times \ell$ matrix whose columns are the $\ell$ atoms in the dictionary. Then we replace the embedding layer with a layer that computes $\Phi\text{softmax}(\Phi^\dagger \nu)$. Effectively, we transform the activation vector $\nu$ into the spectral domain, compute the softmax function on it, and restore the output to activations in the neuron domain. To encourage the network to learn low frequency filters over high frequency we also apply Laplacian smoothing when using a spectral bottleneck.

**MNIST Classifier Architecture**  The basic classifier that we use consists of two convolution and max pooling layers followed by the dense layer where we apply Laplacian smoothing. We use the cross entropy loss to train the classification network in this case. Note that while we use convolutions before this layer for the MNIST example, in principle, techniques applied here could be applied to non image data by using only dense layers until the Laplacian smoothing layer which constructs an

| # | type | patch/stride | depth | output size |
|---|------|--------------|-------|-------------|
| 1 | convolution | 5x5/1 | 32 | 28x28x32 |
| 2 | max pool | 2x2/2 | | 14x14x32 |
| 3 | convolution | 5x5/1 | 64 | 14x14x64 |
| 4 | max pool | 2x2/2 | | 7x7x64 |
| 5 | dense | | 64 | 8x8 |
| 6 | dense | | 10 | 1x10 |

Table 1: Shows the basic MNIST classifier used with and without Laplacian smoothing on layer 5.

| # | type | patch/stride | depth | output size |
|---|------|--------------|-------|-------------|
| 1 | convolution | 5x5/1 | 32 | 28x28x32 |
| 2 | max pool | 2x2/2 | | 14x14x32 |
| 3 | convolution | 5x5/1 | 64 | 14x14x64 |
| 4 | max pool | 2x2/2 | | 7x7x64 |
| 5 | dense | | 64 | 8x8 |
| 6 | convolution | 3x3/1 | 16 | 8x8x16 |
| 7 | max pool | 2x2/2 | | 4x4x16 |
| 8 | convolution | 3x3/1 | 16 | 4x4x16 |
| 9 | max pool | 2x2/2 | | 2x2x16 |
| 10 | convolution | 3x3/1 | 16 | 2x2x16 |
| 11 | dense | | 10 | 1x10 |

Table 2: Shows the MNIST classifier structure with convolutions following the Laplacian smoothing layer (layer 6).

image for each datapoint. Table 1 shows the architecture when no convolutions are used. Table 2 exhibits the architecture when convolution and max pooling layers are used after the Laplacian smoothing layer constructs a 2D image.

