# OpenReview forum: "Graph Spectral Regularization For Neural Network Interpretability"
_ICLR.cc/2019/Conference_

### Official Review · AnonReviewer1 · 2018-10-29
**The usefulness of graph spectral regularizer is shown, but the key points in practice are not considered.**

**Rating:** 4
**Confidence:** 3

**Review:**

Authors highlight the contribution of graph spectral regularizer to the interpretability of neural networks. Specifically, authors consider the Laplacian smoothing regularizer to enhance the local consistency/smoothness between a neuron and its neighbors. Furthermore, by extending the graph Fourier transformation to overcomplete dictionary representation, authors further propose a spectral bottleneck regularizer. Experimental results show that when suitable structural information and corresponding regularizers are imposed, the interpretability of the intermediate layers is improved.

My main concern is that the power of Graph-based regularizer has been well-known in the ML community for a long time. It is not surprising that adding such regularizers to the training process of neural networks can help to get more structural activations. The key points are

1) How to define the Laplacian graph for the neurons? For the simple case shown in Figures 1 and 2, the topology of the neurons has been predefined and the functionality of them is predefined implicitly. For more challenging cases, how to build the Laplacian graph reasonably?

2) How to add the regularizers with good scalability? The complexity of the proposed regularizers is O(N^2) where N is the number of neurons. When the layers contains thousands of neurons or more, how to add the regularizers efficiently?

3) Which regularizer should be selected? Authors propose a class of graph spectral regularizers and their performance is different in different tasks. Is there any strategy helping us to select suitable regularizers for specific tasks?

Unfortunately, authors provide little analysis on these key points.

---

### Official Review · AnonReviewer2 · 2018-11-02
**Latent structure through spectral regularization.**

**Rating:** 3
**Confidence:** 5

**Review:**

The paper introduces a spectral regularization with the aim of obtaining representations
that are easier to interpret.

Some sentences are often confusing and, in general, clarity needs to be improved.

The motivation of the work is not very strong in my opinion, in particular by adding such
a prior the space of possible solutions greatly shrinks and I am afraid
that interesting solutions will be lost. I think one should focus on properties
rather than visual inspection.
Also, isn't it that if we can clearly see the pattern, perhaps that pattern is
linear and of easy discovery also by simpler models?

More importantly, it seems that all experiments are performed on tasks where the
underlying structure is known, however this is almost never the case in practice.
Assuming one uses the proposed spectral regularization, how would one interpret
it in such cases?

In section 2 please clarify the paragraph on bounded Lp norm.

I am sorry but why isn't there a relation, for convolutional nets,
between neurons in different channels? Each element in the feature map represents
the input surrounding that location in a k dimensional space.

The authors state that the usual bottleneck for autoencoders is composed of 2/3
neurons, this is simply not true. There has been extensive work on
overcomplete representations that shows that is better to have many more dimensions
but only few degrees of freedom.

The spectral bottleneck should cite VQVAE as the approach is very similar and the
authors should compare to it.

For the topological inference experiment it is assumed that one knows the structure,
but how to address the more general problem?
More practically, the regularization enforces smoothing (if few eigenfunctions
are used, which is never explained in the paper) between connected nodes, did
the authors try to have a simple L2 penalty instead? E.g. minimize the difference
between activations in the group.

Regarding the capsule network example, when you write that without regularization
each digit responds differently to perturbation of the same dimension, isn't it
possibly true only up to a, unknown, permutation of the neurons?

To summarize, while the idea sounds interesting, I miss to find the easy interpretability
of results and also the overall motivation sounds a bit weak.
More importantly the selection of W, crucial for defining structure, is not discussed at all in the paper.
Experiments are performed on toy examples only whereas here, given that we can
possibly interpret the results I would have liked something more involved to
better show that this kind of interpretability is needed.

Missing cites:
[1] van den Oord et al, Neural Discrete Representation Learning.
[2] Koutnik et al, Evolving neural networks in compressed weight space.

---

### Official Review · AnonReviewer3 · 2018-11-03
**Interesting technique, Lack of Related work**

**Rating:** 4
**Confidence:** 4

**Review:**

Authors present a novel regularizer to impose graph structure upon hidden layers of a neural Network. The intuition is that Neural Networks has typically  symmetric computation among different channels in one layer. Due to the lack of order, visually inspecting the hidden representation is not feasible. By adding edges one can impose a structure upon nodes in one layer and add for example a Laplacian regularizer rather than simple L2 norm regularizer to force the activations to follow the imposed structure.

Pros:

Interesting idea for bringing some benefits of graphical models into Neural Networks using a regularizer.

Experiments verify that one can successfully improve the intrepretability of hidden representations. Also, they provide examples of use cases for such technique like aligning the capsule dimmensions.

Cons:

The major flaw is the lack of comparison with ``any'' of the related work on interpretability or the prior work on imposing structure upon hidden representations. Also, the manuscripts lacks a clear discussion of where does this work stands in the literature like structured VAEs, graphical models, sum product nets + factor graphs.

Also, in none of the experiments authors mention how the added regularizer affects the model performance. Whether imposing the grid structure on CNN (last experiment) drops the CNN accuracy or has no effect? Same for the CapsNet.

Furthermore, the feasibility of calculating the Laplacian for larger scale hidden layers or approximating it is not addressed.

---

### Meta-Review · Area_Chair1 · 2018-12-05
**Structural regularizations imposed on layers.**

**Confidence:** 4
**Recommendation:** Reject

**Metareview:**

The work presents a method of imposing harmonic structural regularizations to layers of a neural network. While the idea is interesting, the reviewers point out multiple issues.

Pros:
+ Interesting method
+ Hidden layer coherence tends to improve

Cons:
- Deficient comparisons to baselines or context with other works.
- Insufficient assessment of impact to model performance.
- Lack of strategy to select regularizers
- Lack of evaluation on more realistic datasets